# Flexible Proteins at the Origin of Life

**DOI:** 10.3390/life7020023

**Published:** 2017-06-05

**Authors:** Andrew Pohorille, Michael A. Wilson, Gareth Shannon

**Affiliations:** 1Exobiology Branch, MS 239-4, NASA Ames Research Center, Moffett Field, CA 94035, USA; michael.a.wilson@nasa.gov (M.A.W.); gareth.b.shannon@nasa.gov (G.S.); 2Department of Pharmaceutical Chemistry, University of California, San Francisco, CA 94132, USA; 3SETI Institute, 189 N Bernardo Ave #200, Mountain View, CA 94043, USA; 4NASA Postdoctoral Program Fellow, NASA Ames Research Center, Moffett Field, CA 94035, USA

**Keywords:** primordial protein structure, flexible protein, ancestral enzyme, ancestral membrane protein, protein ligase, ion channels

## Abstract

Almost all modern proteins possess well-defined, relatively rigid scaffolds that provide structural preorganization for desired functions. Such scaffolds require the sufficient length of a polypeptide chain and extensive evolutionary optimization. How ancestral proteins attained functionality, even though they were most likely markedly smaller than their contemporary descendants, remains a major, unresolved question in the origin of life. On the basis of evidence from experiments and computer simulations, we argue that at least some of the earliest water-soluble and membrane proteins were markedly more flexible than their modern counterparts. As an example, we consider a small, evolved in vitro ligase, based on a novel architecture that may be the archetype of primordial enzymes. The protein does not contain a hydrophobic core or conventional elements of the secondary structure characteristic of modern water-soluble proteins, but instead is built of a flexible, catalytic loop supported by a small hydrophilic core containing zinc atoms. It appears that disorder in the polypeptide chain imparts robustness to mutations in the protein core. Simple ion channels, likely the earliest membrane protein assemblies, could also be quite flexible, but still retain their functionality, again in contrast to their modern descendants. This is demonstrated in the example of antiamoebin, which can serve as a useful model of small peptides forming ancestral ion channels. Common features of the earliest, functional protein architectures discussed here include not only their flexibility, but also a low level of evolutionary optimization and heterogeneity in amino acid composition and, possibly, the type of peptide bonds in the protein backbone.

## 1. Introduction

Proteins mediate most functions of modern cells. A large fraction of cytoplasmic, water-soluble proteins are enzymes that catalyze chemical reactions involved in metabolism and reproduction. Membrane proteins mediate the transport of ions and small molecules across lipid bilayers, as well as essential bioenergetic functions and the transmission of signals from the environment. These proteins are most likely as equally ancient as their cytoplasmic counterparts, and might have evolved separately [1].

Nearly all natural proteins are structurally and functionally complex. The median protein length in eukaryotes has been estimated to be 472 amino acids [2]. Even in simple organisms, proteins are quite large. Their median length in prokaryotes and archaea is approximately 300–320 and 250–280 amino acids, respectively [2,3]. Proteins shorter than 100 amino acids are quite rare. Considering these size distributions, it is highly improbable that functional proteins resembling their modern counterparts in size emerged from an inventory of random polypeptides synthesized genomically or non-genomically at the origin of life. Instead, it is reasonable to assume that contemporary proteins evolved from ancestral forms that were substantially shorter. These ancestors must have been functional; otherwise natural selection would not act in this case. This raises a number of questions. How could small proteins carry out functions of their modern successors, even if less efficiently and selectively? What structural properties characterized ancestral proteins? What were the key evolutionary steps that facilitated their evolution to modern proteins?

Several ideas have been advanced in order to answer these questions [4,5,6,7,8,9]. Although these ideas differ significantly, they share one characteristic—it is postulated that achieving functionality requires a folded, relatively rigid scaffold. At first glance, this assumption is reasonable since evolution has converged on a suite of rigid scaffolds that support a well-defined active site in the case of enzymes and oligomerize to form a stable channel or receptor in the case of membrane proteins [10]. A rigid scaffold pre-organizes the protein and its environment for the desired function. It has been argued that in enzymes this preorganization is largely electrostatic in nature [11]. This type of preorganization is even more evident in ion channels in which the protein is used to create a polar environment for ion transfer through a membrane. To bring about rigid scaffolds, a considerable degree of evolutionary optimization was typically required.

The presence of a well-defined scaffold does not imply that proteins are rigid structures. Some degree of flexibility is required for function. In enzymes, flexibility may support a network of protein conformational motions that increase the rates of enzymatic catalysis [12]. In other instances, flexible loops move to trap the substrate in the active center [13]. In the integral membrane, protein flexibility facilitates transitions between active and inactive states. There are also proteins that are largely intrinsically disordered. In these proteins, conventional elements of a secondary structure often coexist with random coil regions in a loosely packed structure somewhat reminiscent of a molten globule [14,15,16,17,18,19]. They are often involved in translation, transcription, recognition, and cell signaling [20]. Since these functions become more prominent in higher organisms, the fraction of intrinsically disordered proteins increases with organismal complexity [21]. This may indicate that intrinsic disorder is a product of advanced evolution rather than a remnant from ancestral proteins. Occasionally, enzymes can also be intrinsically disordered, although not in parts that contain or support the active center [17].

Here, we argue that at least some of the earliest proteins were considerably more flexible than modern ones and in some instances were built on different principles. Yet, they were functional and the paths to modern, rigid proteins were quite feasible. Dispensing with the requirement for a rigid scaffold markedly increases the suite of potential primordial, functional proteins, because no extensive evolutionary optimization of sequence and structure is required. In support of our argument, we present evidence from experimental studies and our computer simulations that simple, flexible proteins, similar to those that might have existed at the beginnings of life, can efficiently perform functions associated with both water-soluble and membrane proteins.

Here, we focus entirely on the emergence of protein functions and their relation to protein structure. The biochemical aspects of protein synthesis, and in particular the issue of whether the emergence of proteins preceded or followed the emergence of RNA and primitive translational machinery, are not addressed. Additionally, we do not discuss what the primordial inventory of amino acids was and how it evolved. These issues are clearly of fundamental importance to understanding the origins of proteins in general and have a bearing on the origin of protein structure and function, but involve considerations quite different to those addressed here. Readers interested in these subject matters are encouraged to look at excellent reviews and studies of these topics [22,23,24,25,26,27,28,29,30,31,32].

## 2. Water-Soluble Proteins

Almost all modern, water-soluble enzymes contain a well-packed hydrophobic core, which provides structural stability, a largely hydrophilic exterior to maintain water solubility, and a catalytic site. A large fraction of an enzyme is composed of conventional secondary structure elements, α-helices and β-sheets, connected by turns and loops. Based on the universality of these features in modern enzymes, it is usually assumed that their primordial ancestors had the same architecture, but were smaller in size. This assumption, however, creates an evolutionary puzzle. Estimates of the optimal size of proteins characterized by folds with a hydrophobic core and a hydrophilic surface vary between 150 [33] and 200 [34] residues. As we have already pointed out, these structural requirements of modern enzymes are incompatible with the size limitations of primordial enzymes. Some very small proteins can also have a rudimentary hydrophobic core. The 35 amino acid long C-terminus subdomain of actin-binding protein villin is an example of such a protein. It folds to a stable structure in which three α-helices surround a fully dehydrated core consisting of three phenylalanine residues [35,36]. Such small proteins, however, do not exhibit enzymatic activity.

Once proteins reached a sufficient length, several mechanisms were available for their further evolution toward new functions, and improved efficiency and selectivity. Many contemporary, water-soluble proteins rapidly undergo hydrophobic collapse and form native-like elements of a secondary structure, which leads to a molten globule state. In this state, the core is loosely packed and native tertiary contacts produced by the tight packing of amino acid side chains are largely absent. In most cases, this is followed by a transition to a well-ordered, native state. Even though molten globules are considered quite flexible, they can be catalytically active [19,37,38]. It was proposed that they could have preceded modern enzymes before they underwent sufficient evolutionary optimization to adopt fully ordered structures [19]. Molten globules would form a rich reservoir of promiscuous proteins that eventually evolved to specific structures and functions.

A related, but broader, view holds that poorly packed, disordered early proteins exhibited high evolvability [39,40]. According to this view, the ensemble of conformational states available to these poorly optimized proteins contains a number of structures of similar free energies. Thus, upon mutations, such proteins have the potential for both global refolding to a different structure and undergoing local conformational changes near the active center, and by doing so, acquiring new functions.

Although these and related ideas outline plausible evolutionary paths for the functional diversification of proteins long enough to form hydrophobic cores and a sufficient number of secondary structure elements to support refolding, and recognize the significance of protein flexibility to this process, none of them explain how short polypeptide chains were capable of carrying out a variety of catalytic functions. This problem has been well recognized [40]. To resolve this difficulty, several hypotheses have been put forward. One is that precursors of modern enzymes were small, well folded “baby proteins”. Support for this hypothesis is largely drawn from the studies of Baker et al., who successfully designed a number of compact folds of small proteins [5,6]. Most of these proteins, however, do not appear to have any significant enzymatic activity. Alternatively, folded proteins relevant to the origin of life can be obtained from random-sequence libraries by way of in vitro selection and evolution [41,42,43]. This approach was used to obtain a family of small ATP-binding proteins of which the smallest consisted of 45 amino acids [4].

According to another hypothesis, the earliest enzymes were formed from short peptides that relied on self-assembly to adopt ordered structures folded to privileged scaffolds that were capable of supporting a variety of catalytic functions. Several such scaffolds based on α-helices or β-sheets have been tested. A number of them are coil-coils, a common structural motif in modern proteins. It was shown that the aggregation of small peptides to leucine zippers promoted amide bond formation in the templated ligation of two short peptides [7,8]. Several of these systems exhibited properties highly desirable at the origins of life, such as autocatalysis, cross-catalysis, nearly exponential growth, the potential for network formation [44,45,46,47,48], and even chiral selectivity [49]. Hecht and co-workers investigated a number of catalytic peptides based on a four-helical bundle structure [50,51,52], including directed evolution studies [53] and divergent evolution from a generalist to two different specialists [54]. In their recent study [55], with the aid of circular dichroism size-exclusion chromatography and NMR, they found that some of their active, α-helical proteins formed structures that fluctuated between monomeric and dimeric states. This indicates that a fully ordered structure of simple proteins is not a prerequisite for their function. Another coiled-coil scaffold that supports enzymatic function is a 7-helical bundle arranged in a channel-like structure, which has recently been shown to possess strong hydrolytic activity [56].

A different architecture that also relies on the self-assembly of peptides to larger, functional structures is the hydrogen-bonded β-pleated sheet. Many present-day enzymes from *Archaea* have intra-molecularly folded β-solenoid conformations very similar to intermolecularly folded β-amyloid peptides [57]. Even very short peptides can aggregate into such amyloid-like structures, which provide sufficient conformational diversity to create opportunities for many different arrangements of catalytic groups. With only very modest design efforts, it was shown that peptides consisting of only seven amino acids self-assembly to form amyloids exhibiting strong esterase activity [58]. Amyloids are endowed with several advantages. The self-association of peptides into highly stable amyloid structures would partially exclude water from the reaction environment, shifting the equilibrium in the direction of polymer formation. Moreover, amyloids tend to form only between identical or highly similar sequences, often discriminating between peptides of a different length or structure. In a primordial mixture, peptides with unique or closely related sequences of high amyloid-forming ability would self-assemble to form extended β-sheets, whereas unlike peptides would not, thus creating self-reproducing systems. These properties make amyloids attractive candidates for enzymes that might have operated even before the emergence of translational machinery [59]. However, the catalytic potential of amyloids has not yet been characterized. Future work that explores promising clues regarding the ability of amyloids assembled from short peptides to catalyze peptide bond formation, phosphorylation, and specific recognition between amyloids and oligonucleotides can greatly advance our understanding of the role that amyloid-forming peptides might have played in the origin of life.

From a very different perspective, primordial proteins did not have to be well folded to acquire catalytic activity. A rigid scaffold, hydrophobic core, or even traditional elements of secondary structures were not necessary. This greatly relaxes the structural requirements imposed on early enzymes, making their emergence more facile. An example of such a protein was recently discovered through in vitro evolution [60]. This protein catalyzes phosphate bond formation between two fragments of RNA. The protein contains 87 residues, but catalytic activity was demonstrated in markedly shorter variants of this protein in which the tails had been truncated. The efficiency of the ligase is 11^6^–10^7^ above background. Although many natural enzymes exhibit a better efficiency, this catalytic rate is sufficient for many biochemical functions, especially in early cells. For example, the efficiency of the ribosome, which is a formidable synthetic machine, is similar.

The structure of the ligase, later solved by solution NMR [61], contains neither well-defined α-helical or β-sheet regions, nor a hydrophobic core. Instead, it consists of a small, rigid, hydrophilic core containing two small, Zn-binding regions, a flexible loop region which contains the catalytic site, and two flexible tails. In essence, the protein is a flexible, catalytic loop that is tethered at each end by Zn-binding regions. This can be seen in Figure 1a, in which several NMR structures are superimposed and the structure of the protein as a schematic is displayed in Figure 1c. In contrast, the core region is well resolved, as it is essentially the same in all structures. A close-up of the core region in one structure is shown in Figure 1b. The presence of water in the core is supported by Figure 2, which shows a representative snapshot from the molecular dynamics simulations performed and described later in this study. The catalytic site is thought to be located on the C-terminal end of the loop region, as mutations to residues in this region (C47S, E48A, H51A, C53S) greatly reduce or abolish ligase activity.

The stability of the protein was probed experimentally through mutations to structurally important Zn-binding residues. These include (with relative catalytic activity in parentheses): C20S (<25%), C23S (90%), and D34A (~90%), which ligate the N-terminal Zn^2+^ (on the left in Figure 1b), and H18A (50%), E28A (100%), D65A (~90%), C57S (<1%), and C60S(~50%), which ligate the C-terminal Zn^2+^. Additionally, the double mutant E28A/D65A has an activity of ~300% [62]. These results appear to be somewhat counterintuitive. One would expect that mutations to the residues binding zinc should disrupt the structure in the core of the protein and cause the catalytic loop to unravel, which in turn, would abolish catalytic activity. Instead, almost all of the tested mutants were found to retain most, if not full, catalytic activity of the original protein.

To shed some light on the origin of protein robustness with respect to mutations in its core, we carried out molecular dynamics computer simulations of the original ligase and its E28A mutant. The proteins were hydrated with 0.15 M NaCl to match experimental conditions, and production trajectories of 550 ns after equilibration were generated. To assess the flexibility of different parts of the protein, we calculated the root-mean-square deviation (RMSD) of the backbone atoms, which is a measure of how much the structure changes over time relative to the initial structure. We focused, in particular, on residues 18–34 and 57–65, which comprise the two Zn-binding regions, and residues 35–56, which form the loop (see Figure 1a). In accord with the NMR results, we observed that both Zn-binding regions retain their structure, as an RMSD of ~2 Å meant that the core structure remained stable over the course of the simulation (see Figure 3) and did not move away from its initial state to any significant degree. This is in contrast to a rather large RMSD, in the loop region, which indicates that this region undergoes significant structural fluctuations.

The mutation E28A, in which the charged glutamate residue ligating the ion in the C-terminal Zn-binding region is mutated to alanine, was expected to cause disruption to the Zn-binding domain. Surprisingly, the catalytic activity of the mutant was unchanged. Computer simulations showed that the Zn-binding regions remained stable, as the RMSD was in the range 2–2.5 Å for the length of the simulation. Although glutamate in position 28 was no longer available to coordinate zinc, the structural core was maintained with the aid of a neighboring aspartate residue, D29, which ligated the Zn^2+^ ion in place of the mutated E28.

On this basis, one could postulate that a double mutant E28A/D29A should be inactive because it would further disrupt the connection between Zn^2+^ on the C-terminal side of the protein and the N-terminal region, rendering the core unstable. Computer simulations, however, do not support this hypothesis. During a MD trajectory that extended for 2.5 μs, the core underwent some rearrangements, but retained its integrity (see Figure 4). The distance between the zinc atoms did not change appreciably. This is largely due to the short H18-C20 strand, shown in Figure 1c, which constrains the Zn-Zn distance and prevents zinc atoms from separating. The role of this strand in keeping the hydrophilic core intact is supported by the mutation studies presented above, which indicate that substitutions of H18 and C20 with A and S, respectively, decrease enzyme activity. In addition, rearrangements in the core upon the double mutation led to the formation of salt bridges, which further stabilized its structure (see Figure 4). The double mutant was constructed and tested for activity, which was found to be unchanged compared to the original protein (B. Seelig, private communication), in accord with predictions from MD simulations.

The results of the simulations on the E28A and E28A/D29A mutants lead to a suggestion that the robustness of the ligase to mutations arises from small structural rearrangements in the core that are facilitated by the flexibility of the protein. If a residue that binds metal undergoes mutation, this flexibility allows other hydrophilic residues to replace the mutated residue in ligating the ion. Other structural rearrangements provide further stabilization of the core. In this respect, the absence of common elements of a secondary structure is an asset rather than a drawback, as it makes the appropriate rearrangement in the core structure easier to accomplish. The capability to rearrange the core without loss of function increases the number of neutral or near-neutral mutations, which in turn, is beneficial for evolution.

Both experimental and computational results on the ligase illuminate the role of metal ions in supporting the catalytic activity of small proteins. The prebiotic potential of metal ions as catalysts is well established [63,64]. Metal ions also play a structural role in modern proteins, for example, forming the zinc-finger or EF-hand motifs. The occurrence of metal ions in a structural rather than catalytic role has also been observed in a small in vitro evolved protein [4,65]. In the case of the ligase, the organization of the core that prevents the catalytic loop from unraveling is due solely to the ligation of Zn^2+^ by hydrophilic residues. In several modern enzymes, such as rhinoviral protease 2A [66] or NS3 protease from the hepatitis C virus [67], binding zinc induces transitions from a non-functional, intrinsically disordered state to an ordered, active conformation. This suggests that metal ions could turn an inactive, disordered, possibly evolutionarily poorly optimized protein or protein domain into an enzyme. By doing so, they might have formed an evolutionary bridge between proteins without and with a hydrophobic core. Taken together, these results indicate that substituting a hydrophobic core with a rigid hydrophilic core stabilized by way of metal ions provides a solution to an apparent protein size paradox.

The surprising ability of the ligase to retain its functionality in the face of mutations to structurally important residues leads us to propose that similar, small, flexible proteins might have been the evolutionary “missing link” between weakly active oligopeptides [68,69,70] and well-folded enzymes. This hypothesis, as any other regarding the origin of enzymatic activity, requires delineating a plausible path from primordial to modern enzymes. In other words, there must exist a continuous evolutionary path from enzymes containing a catalytic loop attached to a simple hydrophilic scaffold toward increasing complexity. A number of modern enzymes contain catalytic loops. The most notable examples are enzymes based on the TIM barrel fold, which consists of eight–helices and eight–strands alternating along the backbone. Loops that link the helices and the strands at the C-terminal ends tend to contain catalytic sites. Enzymes based on the TIM-barrel fold catalyze a wide variety of reactions, are found in five of six protein classes, and constitute about 10% of all known enzymes [71]. In agreement with a suggestion that TIM barrels facilitated the early evolution of protein-mediated metabolism [72], we suggest that catalytic loops initially supported by a metal-binding hydrophilic core were functional precursors of the TIM scaffold.

## 3. Membrane Proteins

Most modern membrane proteins form multimeric aggregates that contain transmembrane and water-soluble segments. Transmembrane domains have only two types of architectures. A large fraction of them are α-helical bundles. The only other structures are β-barrels. These two architectures are sufficient to carry out a host of different functions that often involve complex, sensitive regulation and high selectivity.

Probably, the initial function of membrane proteins was to act as ion channels that regulated cell volume through equilibrating the osmotic pressure between the protocellular interior and the environment [73,74,75]. Otherwise, cells would either swell or shrink, with highly detrimental effects on their integrity. This implies that the earliest channels might not have been endowed with regulation or selectivity filters, but rather they were just bundles of α-helices that became inserted into the membrane, where they associated to form water-filled pores [1,75]. All that was required from them was to efficiently conduct ions along the concentration gradient.

Lipid membranes are very effective barriers to ion transport. Thus, almost any perturbation to their structure increases ion permeation. Even the simplest mechanisms, such as temperature cycling that creates defects during the liquid/gel phase transition in membranes [76,77] or other types of defects [78], might have increased ion transport rates across the earliest phospholipid membranes. Increased ion permeation due to defects was observed not only in lipid bilayers, but also in protenoid membranes assembled of thermal polymers of amino acids, such as aspartic or glutamic acid [79,80]. However, none of these processes are fast and reliable. Small peptides that were too short to span the lipid bilayer and form channels could have acted to increase permeability. Many of them display antimicrobial activity. They bind strongly to the membrane surface and disrupt the membrane, thus causing indiscriminate leakage [81,82]. RNA molecules that increase membrane permeability act similarly [83]. There is also another mechanism mediated by peptides in the transmembrane orientation that is not disruptive. It relies on the intrinsic flexibility of bilayers, which undergo spontaneous fluctuations in local width, known as capillary waves [84]. These fluctuations, which are frequent but short lived in a pure membrane, can be stabilized by the presence of transmembrane, hydrophobic proteins when their width does not match the hydrophobic core of the membrane. This phenomenon is known as “hydrophobic mismatch”. If a protein is shorter than the average width of the membrane, the bilayer will form a thinning defect around the protein, whereby polar lipid head groups and water penetrate the nonpolar membrane interior. An example of a peptide that induces thinning defects and by doing so increases membrane permeability to ions is trichogin GAIV. This peptide is built of only 10 simple amino acids folded into a helix [85]. Its sequence, Aib-Gly-Leu-Aib-Gly-Gly-Leu_Aib-Gly-Leu, where Aib is α-aminoisobutyric acid, is blocked by n-octanoyl and leucinol at the C- and N-terminus, respectively. Since trichogin spans only half of a typical membrane, it creates deep thinning defects. Trichogin-induced membrane permeability is both ion-selective and dependent the on applied voltage [85]. Short, defect-inducing peptides, such as trichogin, and peptides that could act as ion carriers [86] might have been evolutionary ancestors of membrane-spanning, channel-forming peptides.

It is commonly assumed that a prerequisite for a channel to function is the ability to form a structure that fully encloses a water-filled pore. Even though most modern channels are dynamic in order to gate ions, the transmembrane segment which forms the pore is relatively rigid, as a high degree of flexibility implies a poor stability. In α-helical bundles, the structural integrity of the channel is primarily achieved through strong interhelical, knobs-into-holes packing interactions between neighboring monomers [87,88,89]. These interactions are very clear, for example, in the structures of glycophorin A [90] and the acid-sensing ion channel [91]. In their absence, the channel would open up at the side, completely disintegrate, or collapse, thus closing the pore. In either case, it is presumed that the channel would lose its ability to conduct ions.

Knobs-into-holes interactions that confer the stability and rigidity of channels are most likely a result of extensive evolutionary optimization. This raises a question: how did ion channels appear for the first time? Could they emerge by chance? It would appear that only a small number of amino acid sequences, among those sufficiently long to span the membrane, could assemble to stable, pore forming structures. One hypothesis is that some peptides with very simple sequences were endowed with this property and these peptides were at the roots of modern membrane proteins. An example is a peptide composed entirely of leucine and serine arranged in a heptad repeat sequence [92,93]. This synthetic peptide has been shown to form voltage-gated, hexameric channels. Recent computer simulations indicate that these channels form a remarkably stable coil-coiled structure (Wilson and Pohorille, unpublished).

Alternatively, more frequent, but highly flexible structures that did not fully enclose a pore all the time might have been functional and developed into stable structures through familiar evolutionary processes. So far, however, flexible channels have not been characterized, either experimentally or computationally. The closest, known examples are channels such as alamethicin, that undergo the dynamic exchange of helices, as proposed in the “barrel stave” model [94,95]. In addition, alamethicin appears to exhibit considerable structural heterogeneity, as revealed in combined coarse grained and atomistic molecular dynamics simulations of this channel in a lipid bilayer on a time scale of microseconds [96]. A similar behavior was observed in atomistic molecular dynamics simulations (Wei and Pohorille, unpublished) of the tethered alamethicin hexamer [97]. 

In search of potentially flexible models of ancestral ion channels, we turn to peptaibols, a family of naturally occurring antimicrobial peptides from fungi, and in particular to antiamoebin (AAM) [98,99]. Although peptaibols are not ancient, antiamoebin is a particularly attractive model of the earliest ion channels. It is one of the shortest channel-forming peptides known, consisting of only 16 amino acids, and exists as an α-helix in a nonpolar environment. The sequence of amino acids is Ac-Phe-Aib-Aib-Aib-Iva-Gly-Leu-Aib-Aib-Hyp-Gln-Iva-Hyp-Aib-Hyp-Aib-Pro-Phol, where Aib, Iva, Hyp, and Phol stand for α-aminoisobutyric acid, isovaline, hydroxyproline, and phenylalinol, respectively. None of the residues in AAM are charged. The peptide is believed to also act as an ion carrier [86]. This might point to an ancestral relation between channel-assisted and carrier-assisted ion transport. Non-standard amino acids in AAM, such as α-aminoisobutyric acid and isovaline, prevent bacterial proteases from recognizing and degrading the peptide. Interestingly, the same amino acids have been found in meteorites and are assumed to have been common on the early earth [100]. Since AAM is synthesized non-ribosomally and, therefore, is not genomically coded, it has not been subjected to common, highly effective evolutionary optimization whereby single-point mutation changes the amino acid sequence and modifications that improve fitness are retained in the genome of subsequent generations.

Voltage gated AAM channels were studied electrophysiologically [86]. In most measurements, the channel exhibited a single conductance level, indicating the presence of only one structure. This is in contrast to alamethicin, also a peptaibol, which has multiple conductance levels, presumably associated with multiple structures, each containing a different number of helices [94]. Single channel recordings carried out for AAM yield a conductance of 90 pS at an applied voltage of 75 mV, adjusted to an ionic strength of 1 M KCl [86]. Measurements of macroscopic conductance as a function of peptide concentration led to the conclusion that the number of helices forming the channel is a multiplicity of three or four [86]. Computational studies in which a large number of tetrameric, hexameric, and octameric structures were considered led to the conclusion that the AAM channel is a hexamer [101]. All tetrameric structures were found to be non-conducting, whereas stable octameric models exhibited conductance significantly higher than the experimental value.

Considering that the AAM peptide is quite short, contains almost entirely simple amino acids, and has had only limited opportunities for evolutionary optimization, we hypothesized that the AAM channel might be more flexible than most other ion channels, even though such behavior was not apparent from a previously generated MD trajectory 150 ns in length [101]. To test this hypothesis, we extended simulations of the AAM hexamer in the palmitoyloleoylphosphatidylcholine (POPC) bilayer to microsecond time scales. Three trajectories, abbreviated T1, T2, and T3, were generated. As described in the Methods section, the initial channel structure in each run was different and intermolecular interactions were described by way of different model potentials. The MD trajectories for these three systems exhibited similar features. On a time scale of several microseconds, the channel underwent very large fluctuations. The structures lost their initial symmetry relatively quickly. In a number of instances, one helix appeared to be in the process of dissociating from the bundle, but it always remained attached to the rest of the assembly. Occasionally, the structures opened at the side, allowing for direct interactions between water molecules filling the pore and phospholipids forming the membrane. These openings, however, were subsequently sealed. Sometimes the channel closed almost completely, leaving only a small number of water molecules in the pore, but it opened up again, as the simulations continued. This is shown in Figure 5 and in two movies, Appendix A. The key to preserving the structural integrity of this largely hydrophobic channel is the formation of intermolecular interactions between glutamines, which are located near the middle of the monomers, a feature shared with alamethicin [102], although interactions between other residues also contribute to the channel stability.

In all three simulations, the channel conducted ions at highly non-uniform rates. Periods in which ion currents were large, small, and absent were observed. The currents correlated well with the number of water molecules in the pore. Currents were large when the water-filled pore was also large, and decreased as the number of water molecules in the pore decreased. For T1 and T2, these currents were similar, but followed each other in different orders. This can be seen in Figure 6, in which the total number of K^+^ and Cl^−^ ions that cross the channel is shown as a function of time. Considering these similarities, one can pool the results from both runs to obtain an estimate of channel conductance. Since periods of high and low conductance are long relative to the total length of the trajectories, this estimate is burdened with substantial uncertainty. In such circumstances, the best approach to estimating the conductance is by way of a bootstrapping method. This yields 90 ± 23 pS at 75 mV, which agrees well with the measured conductance. This value is similar to the conductance of a number of other small channels. For example, the conductance of a presumably hexameric channel formed by a 30 residue long transmembrane fragment of Vpu, a membrane protein encoded by the HIV-1 genome, is equal to 96 pS in the 1,2-Dioleoyl-sn-glycero-3-phosphocholine membrane in the presence of 0.5 M KCl and an applied voltage of 50 mV [103]. In general, biological channels exhibit a very broad range of conductance values. For low conductance channels, such as acid-sensing ion channels, these values are 10–15 pS [104], whereas for high conductance channels, such as MscL and α-hemolysin, they exceed 1 nS [105,106]. The conductance of one of the most extensively studied ion channels, the voltage gated potassium channel KcsA [107], has been measured to be in the range of 80–160 pS at 100 mV, adjusted to 200 mM of KCl [108,109,110].

The conductance pattern is somewhat different in T3. The water pore was quite large in this case, which resulted in a markedly higher conductance. Nevertheless, even for this trajectory, the channel remained quite flexible and conducting for nearly four microseconds without disintegrating.

One might be concerned about whether the observed high flexibility of the channel could be an artifact of computer simulations due to incomplete equilibration or inaccuracies in potential functions. In other words, a rigid structure of the AAM channel might exist, but was never found in our simulations. Although this possibility cannot be unambiguously excluded, several arguments indicate that this is not the case. First, it might appear that the channel simply undergoes the transition from a low-activity to a high-activity state and then stays there for the remainder of the simulations. This is, however, not the case. Transitions in the opposite direction also occur, as can be seen for T1. Second, different simulations were started from different structures. During the course of the MD trajectories, many structures of the channel were explored, but apparently none of them were very stable. In particular, T2 was initially set as a left-handed coiled-coil, but ended in a distorted, right-handed coil-coil. Many intermediate structures were explored in the process, but all of them were quite flexible. Furthermore, the simulations were carried out using different force fields. Yet, the behavior of the channel was qualitatively similar. The CHARMM force field was recently used to describe another small ion channel—the M2 proton channel of the Influenza A virus—on very similar time scales and yielded perfectly stable structures that fluctuated around their equilibrium positions with only small RMSD [102]. A movie, Appendix A, with the MD trajectory for this channel, is given in the Appendix A for comparison with Appendix A. Taken together, these arguments suggest that the large fluctuations of the AAM channel observed in the MD simulations are not artifacts, but rather are characteristic of the actual behavior of this channel. This view has indirect support from experiments. In comparison to most other channels, measurements of the current through the AAM channel are difficult and conductance depends on sample preparation [86], which suggest that the channel is fragile.

The results of our simulations are consistent with a picture in which the earliest ion channels were loose bundles of α-helices that had to contain at least one polar residue to keep the structure together. The specific identity of amino acids in a helix appears to be less important, a desirable trait in early evolution. The presence of simple, non-standard amino acids that were common in the primordial environment does not prevent the channel from being functional. Moreover, there is no requirement for amino acids that are synthesized in complex biosynthetic pathways and are assumed to emerge late in protobiological evolution [28,29]. Taken together, this means that simple ion channels might have existed even before the precise protein synthesis mechanisms or the full suite of amino acids were present, thus protecting nascent cells from osmotic disequilibria. As a consequence of rather minimal structural requirements, the channels were highly flexible, yet they remained functional. They formed the “feed stock” for the subsequent evolution to modern channels.

## 4. Methods

### 4.1. Molecular Dynamics Simulations of Water-Soluble Proteins

The system contained the ligase protein, 87 residues long (1301 atoms), 29,029 water molecules, two Zn^2+^ ions bound in the core of the protein, 85 Na^+^ ions, and 80 Cl^−^ ions. This corresponds to a salt concentration of 0.15 M to match the experimental conditions. For the E28A mutant, only 84 Na^+^ ions were used in the simulation, as a neutral alaninie was substituted for a negatively charged glutamate residue in the original protein. The initial structure of the E28A mutant was obtained by way of homology modeling with respect to the NMR structure of the original protein [111].

The initial system was constructed by placing either the best NMR structure or the E28A mutant in an equilibrated box containing 30,000 water molecules, removing overlapping water molecules, and mutating 165 water molecules at random to Na^+^ and Cl^−^ ions. Any initial close contacts were removed by minimizing the positions of the water and ions, while keeping the protein rigid. A trajectory of 20 ns was then constructed to relax the water and ions while the protein was held rigid. Then, only the protein backbone was held fixed and an additional 20 ns trajectory was generated. Finally, all restraints were released and an additional trajectory of 50 ns was used to equilibrate the system. Finally, production runs of 550 ns were generated. The average size of the simulation cell was 95.8 Å on a side (the box was only allowed to fluctuate uniformly).

Periodic boundary conditions were applied in all three spatial directions. The equations of motion were integrated using a 2 fs time steps. Particle Mesh Ewald was applied to treat long-ranged electrostatic effects. All simulations were carried out at a constant temperature of 300 K using Berendsen thermostats, and a constant pressure of 1 atm using MTK barostats. The initial setup and equilibration was carried out on the Pleiades supercomputer at NASA Ames. The final 50 ns equilibration and 550 ns production runs were carried out on the Anton supercomputer at the Pittsburgh Supercomputer Center. The protein and ions were modeled using the CHARMM22/CMAP force fields [112,113].

### 4.2. Molecular Dynamics Simulations of Antiamoebin

The system under study consisted of an AAM ion channel embedded in a POPC bilayer placed between water lamellae. The system was enclosed in a rectangular simulation box with periodic boundary conditions applied in all three spatial directions. The size of the box was chosen such that the calculated surface coverage per phospholipid molecule away from the peptide matched the experimental value [114], and the calculated density of water away from the bilayer was equal to the density of bulk water.

Channel models were built of six monomers and equilibrated in the membrane using the procedures described before [101]. The initial structures for T1 and T2 were the structures studied previously [101] and T3 was a left-handed coiled-coil. MD simulations were carried out in the NVT ensemble. Equations of motion were integrated using a 2 fs time step. Particle Mesh Ewald was applied to treat long-ranged electrostatic effects. The temperature was kept constant with the aid of the Berendsen thermostat. An electric field was applied in the z-direction perpendicular to the interface. The applied voltage was equal to 150 mV, which is twice the value used in most experiments. This was done to improve the statistics for the calculated current and provide a direct comparison with the previous study [101]. All simulations were carried out on the Anton supercomputer. Other information about the simulated system, such as system sizes, force fields used, lengths of MD trajectories, applied voltages, and the number of ion crossing events, is given in Table 1.

The single-channel conductance was calculated as the ratio of the observed current through the channel to the applied voltage. The current was estimated by counting the number of ions that cross the channel during the simulation. More accurate methods for estimating currents are also available [115]. In our previous work on AAM, they were shown to yield the same conductance as direct ion counting to within statistical errors.

## 5. Summary and Conclusions

It is commonly assumed that primordial proteins had ordered scaffolds that provided preorganization for activity [55]. These scaffolds were built of structural elements that shared a number of properties with modern proteins. This point of view has guided most efforts to construct models of the earliest functional proteins. Several interesting ideas have been put forward to explain how function can be achieved through the self-assembly of small peptides into specific, ordered scaffolds. In variance with these ideas, it was recently shown that structurally dynamic, simple proteins that are compatible with the conditions at the origin of life could be functional in vivo [55]. Here, we significantly extend this notion. We present results from experiments and our computer simulations indicating that a rigid scaffold was not a prerequisite for either primordial water-soluble enzymes or membrane channels. These two types of proteins carried out functions that were ubiquitous for the nascent cellular life. Instead, the earliest functional proteins might have been quite flexible. Such proteins did not require extensive evolutionary optimization of their sequence to achieve activity, a desirable property that might have considerably increased the repertoire of functional polymers at the origins of life. In particular, it has been determined through in vitro evolution experiments that a model of ancestral, water-soluble enzymes has an architecture quite different from what is found in biological systems. Contemporary enzymes form a hydrophobic core, which requires polypeptide chains of a considerable length. The model protein, instead, has a small, rigid, hydrophilic core stabilized with the aid of metal ions that tether the ends of a highly flexible catalytic loop. This markedly reduces the length requirements for enzymatic activity, as it is no longer needed to surround a hydrophobic core with a hydrophilic exterior exposed to water. Common elements of the secondary structure, such as α-helices and β-sheets, are neither required nor found in this novel architecture. This appears to impart considerable robustness to mutations in the core, even if they affect critical interactions with the metal ligands, as the polypeptide chain not engaged in an ordered secondary structure readily undergoes rearrangements to accommodate mutations. 

Flexible loops in the active center are common in modern enzymes. We hypothesize that as early enzymes evolved toward longer chains, hydrophilic cores were substituted by hydrophobic cores that were more stable and better pre-organized for catalytic activity. Proteins based on a frequent and ancient TIM barrel architecture might be examples of such enzymes. The transition from a hydrophilic to a hydrophobic core would require the global structural reorganization of a protein. Experimental data and theoretical considerations suggest an evolutionary mechanism that might lead to such reorganization [40,116].

The ancestral membrane proteins also did not have to be rigid to maintain their earliest functions, such as equilibrating the osmotic pressure across protocellular walls. Even very short peptides that could barely span the membrane could act as ion carriers and form ion channels upon aggregation. Very limited evolutionary optimization was required, as could be ascertained from the very simple sequence of AAM. The consequence of this simplicity and poor evolutionary optimization was high flexibility, which implies weak stability, weak ion selectivity, and the absence of gating or regulation. Only later in evolution, the stability of channels increased through optimized, knobs-into-holes interhelical interactions.

Flexible proteins might not have been the only ones that existed at the early stages of protocellular evolution. Catalytic, amyloid-like assemblies and leucine/serine ion channels are promising models of rigid, functional peptide structures. Another example is a small, ATP-binding protein evolved in vitro from a library of random amino acid sequences [4]. This protein exhibits a well-defined, novel fold stabilized by a zinc atom. It is quite possible that both rigid and flexible proteins contributed to the pool of functional proteins at the origin of life.

Differences between primordial and modern proteins can also be viewed from a broader perspective. High flexibility, in some cases based on different structural principles, was only one feature that distinguished ancient proteins from their contemporary descendants. There might have been other differences. Polypeptide chains might have contained amino acids that were synthesized abiotically and existed on the early Earth, but are not in the suite of 20 canonical amino acids. In fact, some of them are still present in simple, non-genomic peptides and have no negative impact on their functions. Even though non-canonical, abiotically synthesized amino acids were not as complex and functionally versatile as the so-called “late”, canonical amino acids, such as histidine or phenylalanine, their presence in early proteins could broaden and improve the activity of these proteins beyond what would be available if only the “early” canonical amino acids were used. Further, it appears that the existence of proteins containing only α peptide bonds, as is the case in modern proteins, is not a requisite for function. Since the formation of polypeptides with all α bonds is often considered unlikely in the absence of ribosomal-like machinery, this requirement has been sometimes taken as evidence for the translational origins of the first proteins [117]. Yet there is no obvious reason to assume that the presence of some β bonds would render enzymes with a hydrophilic core inactive. The same applies to proteins with a hydrophobic core, as it has been long established that many different, water-soluble polymers would tend to collapse to three-dimensional structures similar to that adopted in modern proteins, driven by hydrophobic interactions [9]. Considering similarities in side chains, such structures could have been functional, even if the polypeptide chain did not contain all α bonds. 

In summary, it appears that rigid scaffolds, the presence of only canonical amino acids, and the α-polypeptide backbone were not required for function. This means that the earliest proteins might have been considerably more heterogeneous in sequence and structure than frequently assumed. Furthermore, reconstructing the early evolutionary history of functional proteins might be very challenging, as a considerable part of it was erased by modern cellular machinery. For this reason, standard techniques of phylogenetic reconstruction, which have been extremely successful for tracing subsequent protein evolution, are unlikely to be useful. Instead, in order to explore the functional potential of the earliest, heterogeneous proteins, methods of biochemistry and molecular biology, such as in vitro or in vivo evolution, have to be considered.

## Figures and Tables

**Figure 1 life-07-00023-f001:**
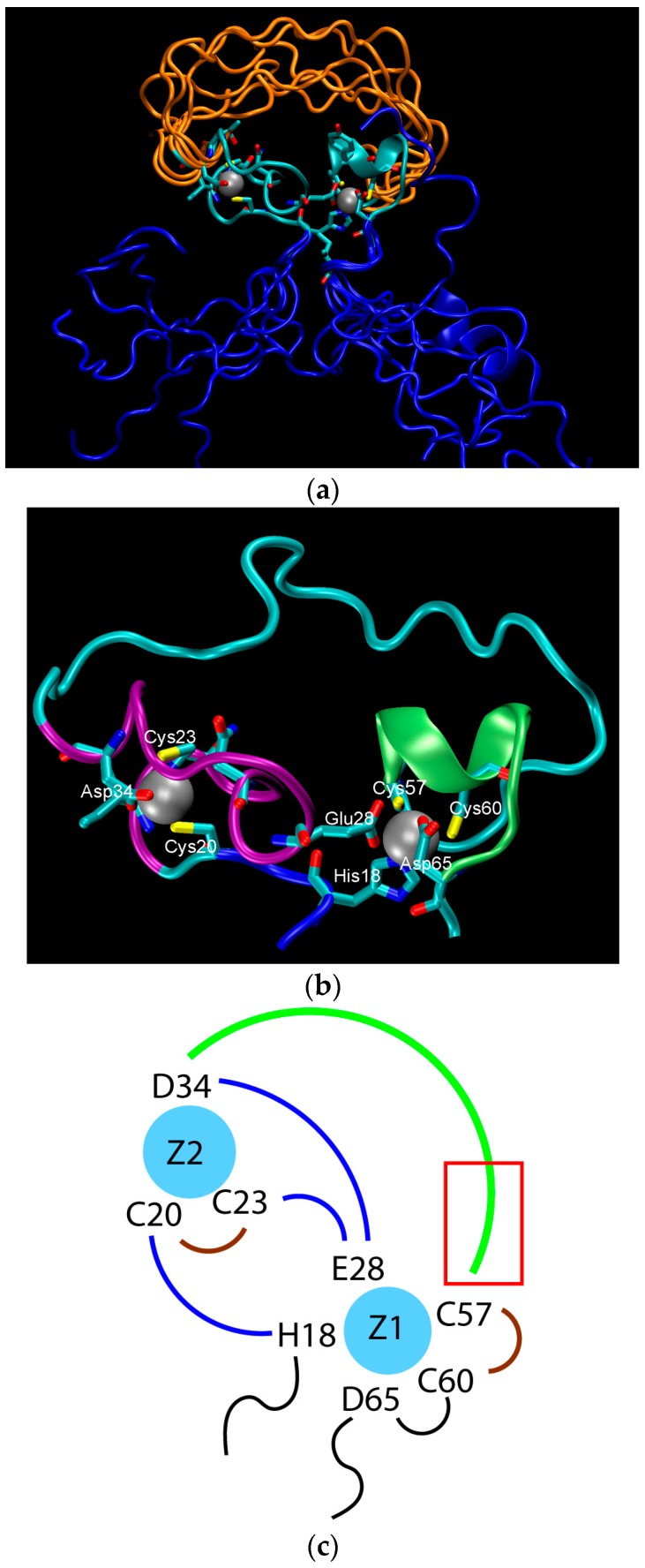
The structure of the ligase discovered through in vitro evolution [61]. (**a**) Superimposed NMR structures demonstrating the flexibility catalytic loop in orange, the rigid hydrophilic core in gray, and the flexible termini in blue; (**b**) The zinc-binding residues of the rigid hydrophilic core. Zinc atoms are grey balls. Residues coordinating them are marked; (**c**) Schematic of the zinc-binding residues. Strands that connect zinc atoms are in blue. The residues considered to be involved in coordinating zinc atoms are represented explicitly. The catalytic loop is in green. The red rectangle indicates the region presumed to contain catalytic residues.

**Figure 2 life-07-00023-f002:**
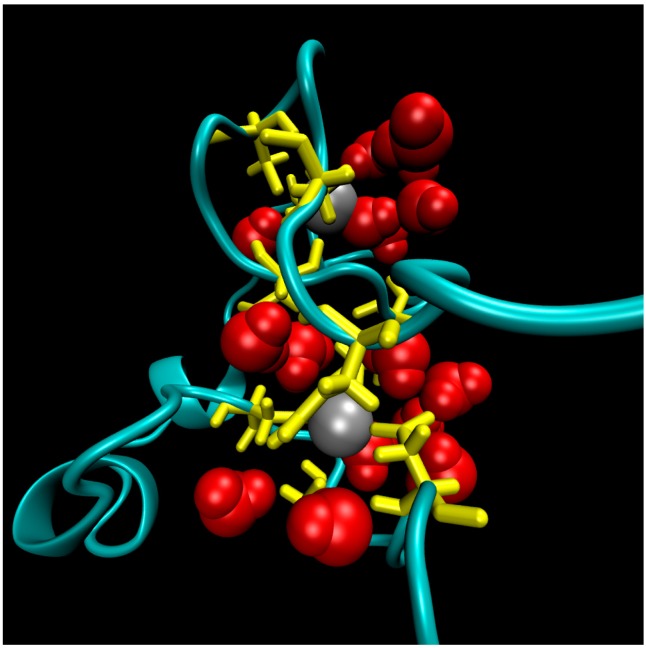
A representative snapshot of the hydrophilic core of the ligase from Molecular Dynamics simulations. The point of view is that of the flexible termini. Protein residues in the core are colored yellow, zinc ions atoms are grey balls, and water molecules within 5 Å of zinc ions are red.

**Figure 3 life-07-00023-f003:**
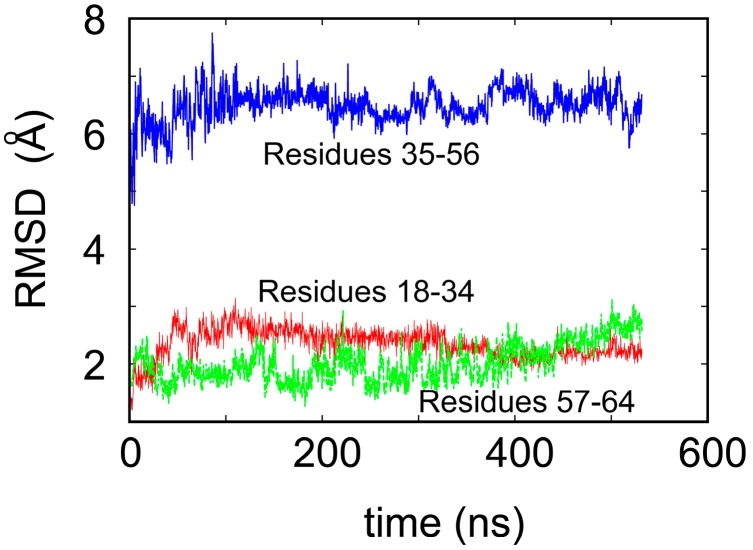
Root-mean-square deviation (RMSD) of the backbone atoms of the ligase over 550 ns of the molecular dynamics simulation.

**Figure 4 life-07-00023-f004:**
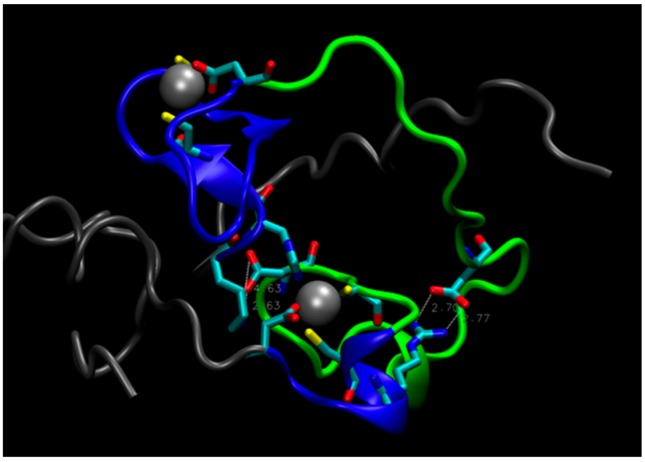
The structure of the double mutant E28A/D29A. Tails (residues 1–17 and 66–87) are gray, loop (residues 35 to 59) is green, residues 18–34 and 60–65 near Zn atoms are blue. White, dashed lines indicate salt bridges Asp55-Lys17 (on the left) and Asp43-Arg61 (on the right).

**Figure 5 life-07-00023-f005:**
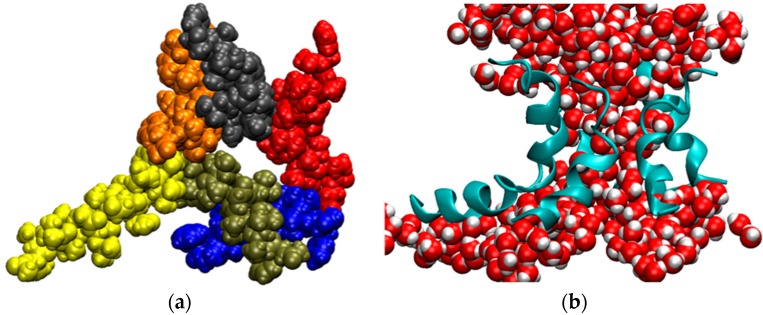
Snapshot from a simulation of the AAM channel after 3.6 μs (trajectory T2). (**a**) View of the channel from above. Each monomer of the hexametric structure is in a different color; (**b**) Side view of the channel represented as ribbons and water molecules filling the pore. All other components of the system were removed for clarity.

**Figure 6 life-07-00023-f006:**
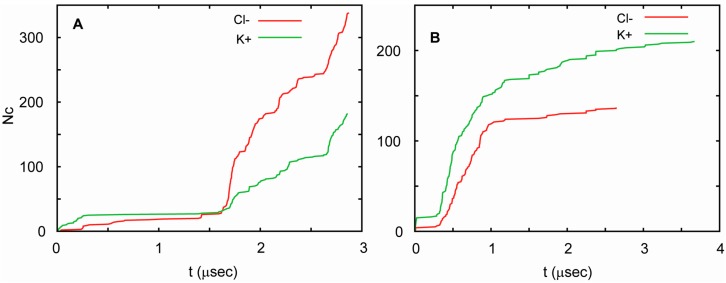
Cumulative transmembrane ion fluxes in simulations of AAM. The total number of Cl^−^ ions (red) and K^+^ ions (green) that cross the channel as a function of time for trajectory T1 (**A**) and T2 (**B**). The simulations were carried out at an ionic strength of 1 M (see Table 1).

**Table 1 life-07-00023-t001:** Details of the MD simulations presented in this study.

Trajectory	T1	T2	T3
Simulation time (µs)	3.0	3.8	3.8
Box size x, y, z (Å)	86.3, 84.1, 93.9	82.3, 84.3, 108.2	83.66, 83.66, 106.62
No. of lipid/water molecules	222/13053	200/16228	198/16228
Protein potentials	CHARMM22	AMBER99	CHARMM22
Lipid potentials	CHARMM27	CHARMM36	CHARMM36
Water potentials	TIP3P	SPC/E	TIP3P
Ionic strength	1 M	1 M	1 M
Voltage, V (mV)	150	150	150

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
