# Peer review of "Flexible Proteins at the Origin of Life"

_life, 2017, doi:10.3390/life7020023_

Round 1

Reviewer 1 Report

This is an interesting article, however, there are some major problems that prevent its publication.

Do the authors intend their manuscript to be (a) a research article, (b) a review article, or (c) a new “Hypothesis in the Life Sciences” ?   (Such hypotheses papers are described in this mdpi website:   https://www.mdpi.com/journal/life/sections/hypotheses   Unfortunately, the manuscript does not succeed in any of these three categories.  It falls short as a Research Article because it barely reports any new results.  The authors did a few MD simulations. (2 figures in a 22 page manuscript.)  MD is a mature field and I’m not sure that two MD simulations justify a full research paper. This manuscript is also not a Review Article.  It does not fully review either of the broad topics “flexible proteins” or “origin of life.”  Nor does not adequately review previous work on the narrower topic of the title “Flexible Proteins at the Origin of Life.”  Finally, the paper does not present a new “Hypothesis in the Life Sciences.”  The concept that protein flexibility may have been important in the origin of life has been discussed previously in several publications.  A Google search of the title of this paper “Flexible Proteins at the Origin of Life” (and a few related searches) yielded many hits.  Some of these earlier works explicitly discuss protein flexibility in designed proteins that may mimic proteins from early evolution.  The authors seem unaware of previous work in this field.  Some of this previous work is summarized in the links below:

https://www.sciencedaily.com/releases/2016/01/160106215533.htm

https://news.illinois.edu/blog/view/6367/204758

http://www.astrobio.net/also-in-news/mutation-triggered-multicellular-life-altered-protein-flexibility/

http://www.sciencedirect.com/science/article/pii/S0014579398006747

https://www.ncbi.nlm.nih.gov/pubmed/26707197

https://www.ncbi.nlm.nih.gov/pubmed/27618189

http://www.nature.com/articles/srep44948

https://arxiv.org/abs/1212.2822

http://news.emory.edu/stories/2016/01/ortlund_ancient_protein_flexibility_pnas/index.html

http://www.pnas.org/content/101/35/12860.full

Beyond the major issues mentioned above, there are also a few minor issues

-- The first sentence of the introduction is repeated twice.

-- Line 431 states “The results of our simulations lead to a picture in which the earliest ion channels were loose bundles of α-helices that….”  How can a simulation of an existing protein lead to a picture about the earliest ionic channels?   While it is true that results of the simulation are consistent with “a picture in which…” they do not lead to this picture.

-- Line 444 states “In most hypotheses about the origin of protein function, it is assumed that primordial proteins had ordered scaffolds that provided preorganization for activity…”   This referee is is not aware of such hypotheses.   Is there a reference for this? 

-- Line 448 states “Here, we present evidence from experiments and our computer simulations that rigid scaffolds were not a prerequisite for either primordial water--soluble enzymes or ….”  Previous evidence for this claim was published by Hilvert’s lab and should be referenced.  See http://www.pnas.org/content/101/35/12860.full

Reviewer 2 Report

p.p1 {margin: 0.0px 0.0px 8.0px 0.0px; font: 11.0px Helvetica} p.p2 {margin: 0.0px 0.0px 8.0px 0.0px; font: 11.0px Helvetica; min-height: 13.0px} li.li1 {margin: 0.0px 0.0px 8.0px 0.0px; font: 11.0px Helvetica} span.Apple-tab-span {white-space:pre} ol.ol1 {list-style-type: decimal}

Wilson and coworkers present MD simulations for two model systems:

1.  An in vitro evolved ligase, in which mutation of a key, zinc-chelating Glu residue can be compensated for by a neighboring Asp residue by virtue of the flexibility of the backbone (already reported, though not analyzed by MD).

2. Antiamoebin, which is taken to be an example of a primitive ion channel and is, apparently, quite flexible by MD standards.

On the basis of these results – and an analysis of the literature – the authors believe they have provided support for the hypothesis that “rigid scaffolds were not a prerequisite for either primordial water-soluble enzymes of membrane channels,” among other things.  My guess is that the MD simulations reported here will not be particularly interesting to most researchers in the field and absolutely do not command a title as strong as “Flexible Proteins at the Origin of Life.”  I agree with the authors that highly flexible proteins are an important resource to understand the properties of the first proteins, and may have been a potential precursor to modern-day folded proteins – but these ideas have already been proposed, and much more convincingly so (for example, see PMID:  27623012, which is not cited).  The manuscript was clearly intended as a review (with over 100 references, and a broadly imagined title) and, in my opinion, should be returned to that state (by removing a bulk of the MD analysis).  Otherwise, the manuscript should be re-written as a proper research article that focuses more on framing the specific, novel results presented by the authors.

Minor comment:

--The first sentence is repeated.

Round 2

Reviewer 1 Report

no comments